# Usability of myfood24 Healthcare and Mathematical Diet Optimisation in Clinical Populations: A Pilot Feasibility Randomised Controlled Trial

**DOI:** 10.3390/nu14091768

**Published:** 2022-04-23

**Authors:** Diane E. Threapleton, Sarah L. Beer, Dustin J. Foley, Lauren E. Gibson, Sarah Trevillion, Dermot Burke, Pete Wheatstone, Jacqui Gath, Nick Hex, Jo Setters, Darren C. Greenwood, Janet E. Cade

**Affiliations:** 1Nutritional Epidemiology Group, School of Food Science & Nutrition, University of Leeds, Leeds LS2 9TJ, UK; dianethreapleton@outlook.com (D.E.T.); d.j.foley2@leeds.ac.uk (D.J.F.); wheatpd@yahoo.co.uk (P.W.); jgath@blueyonder.co.uk (J.G.); 2Dietary Assessment Ltd., Nexus Building, Discovery Way, Leeds LS2 3AA, UK; s.l.beer@myfood24.org (S.L.B.); l.e.gibson@myfood24.org (L.E.G.); 3Leeds Institute for Data Analytics, University of Leeds, Leeds LS2 9NL, UK; d.c.greenwood@leeds.ac.uk; 4Tier 3 Weight Management Team, York and Scarborough Teaching Hospitals NHS Foundation Trust, Wigginton Road, York YO31 8HE, UK; sarah.trevillion@york.nhs.uk; 5LIMR Division of Gastroenterology and Surgery, University of Leeds, St James’s Hospital, Leeds LS9 7TF, UK; d.burke@leeds.ac.uk; 6York Health Economics Consortium, University of York, York YO10 5NQ, UK; nick.hex@york.ac.uk (N.H.); jo.setters@red-al.com (J.S.); 7School of Medicine, University of Leeds, Leeds LS2 9NL, UK

**Keywords:** nutrition app, myfood24, dietary assessment, smartphone, feasibility, randomized controlled trial

## Abstract

Monitoring nutritional intake is of clinical value, but few existing tools offer electronic dietary recording, instant nutritional analysis, and a platform connecting healthcare teams with patients that provides timely, personalised support. This feasibility randomised controlled trial tests the usability of ‘myfood24 Healthcare’, a dietary assessment app and healthcare professional website, in two clinical populations. Patients were recruited from a weight management programme (n21) and from a group of gastroenterology surgery outpatients (*n* = 27). They were randomised into three groups: standard care, myfood24, or myfood24 + diet optimisation (automated suggestions for dietary improvement). The participants were asked to record their diet at least four times over eight weeks. During the study, healthcare professionals viewed recorded dietary information to facilitate discussions about diet and nutritional targets. The participants provided feedback on usability and acceptability. A total of 48 patients were recruited, and 16 were randomised to each of the three groups. Compliance among app users (*n* = 32) was reasonable, with 25 (78%) using it at least once and 16 (50%) recording intake for four days or more. Among users, the mean (standard deviation) number of days used was 14.0 (17.5), and the median (interquartile range) was six (2.5–17.0) over 2 months. Feedback questionnaires were completed by only 23 of 46 participants (50%). The mean System Usability Score (*n* = 16) was 59 (95% confidence interval, 48–70). Patient and healthcare professional feedback indicates a need for more user training and the improvement of some key app features such as the food search function. This feasibility study shows that myfood24 Healthcare is acceptable for patients and healthcare professionals. These data will inform app refinements and its application in a larger clinical effectiveness trial.

## 1. Introduction

Many patient populations and clinical consultations could benefit from a structured assessment of dietary habits and personalised support. One in five early deaths worldwide is associated with poor dietary habits [1]. Diet is an important factor in the health status or recovery in many patient groups, and nutritional therapy, with the assessment of nutritional status, is strongly recommended before and after surgery [2]. A systematic review of dietary mobile apps and nutritional outcomes in adults with chronic diseases concluded that apps were effective self-monitoring tools, and that their use resulted in positive effects on measured nutritional outcomes [3]. Over 90% of individuals with cancer want advice on diet, activity, and weight [4]. Cancer patients want to be ’treated as individuals, with individual cancers’, suggesting that personalised advice would be well-received. Furthermore, among weight loss patients, having a strong relationship with a healthcare professional was identified as the main driver for successful long-term weight loss with the help of e-health tools [5]. Barriers to assessing diet and providing personalised nutritional support include overall cost, the labour-intensive nature of reviewing recorded dietary intake, and the varying needs within any patient population. Technology can offer solutions to address such barriers, and many applications (apps) have been developed to support behaviour change. However, few additionally provide a direct platform for Health Care Professional (HCP) involvement in the care of clinical populations, and those that do are not backed by scientific data from validation studies or clinical trials. 

### 1.1. Improved Health and Recovery through Technology-Supported Dietary Behaviour Change

Technology is needed to support dietary assessment in clinical settings, where the tracking and self-monitoring of food and nutritional intake may help treatment and recovery and be less burdensome for clinical input. For example, malnutrition is common among gastroenterology patients and can slow down recovery [2]. Additionally, routine screenings with 3 × 24 h dietary recalls can effectively improve energy intake and preserve body weight among oncology patients [6]. A systematic review of studies examining the role of nutritional status in quality of life (QoL) among cancer patients also found that nutrient status is a strong predictor of QoL [7]. Therefore, correcting malnutrition in vulnerable patient groups may improve a range of outcomes, including QoL, an important outcome for both patients and families. Improved nutrition behaviours and nutrition-related health outcomes, including those related to obesity, blood pressure, and blood lipids, were also reported from app-based nutrition interventions that employed four common behaviour change techniques, namely ‘goals/planning’, ‘feedback/monitoring’, ‘shaping knowledge’, and ‘social support’ [8]. 

The adoption of mobile phones and devices has created an opportunity for assessing and improving nutrient intake. A review of 13 of the most popular (downloaded) Android and iPhone diet apps indicates that there is a clear interest and opportunity for diet monitoring and for providing recommendations, but none of the apps evaluated were capable of providing personalized diet feedback [9]. A more recent review of the top Android and iPhone diet-tracking apps identified their use by the general population and in very broad markets (ranging from hundreds of thousands to millions of app downloads), but it noted that they did not provide personalised feedback beyond nutritional summaries of foods consumed [10]. Existing consumer fitness or weight-loss apps provide generic online solutions for healthy populations but are limited in terms of (i) having no integration with healthcare information technology (IT) systems that support professional–patient communication [11], (ii) incomplete food and drink datasets, (iii) output nutrients not providing the full clinical nutrition picture [11], (iv) having no validation against independent biomarkers of nutrients that demonstrate the accuracy of the measures [12], (v) providing misleading or inaccurate information [13], and (vii) having no guidance supporting behaviour change [9]. Healthcare professionals need apps which have been collaboratively designed to take account of their current practise in order to enhance patient care [14].

### 1.2. myfood24 Healthcare Technology

Dietary behaviour change strategies include goal setting, personalised care, HCP support, and self-monitoring [8], all of which are facilitated through the myfood24 Healthcare platform. The usability of myfood24 has been assessed in the general population [15], but the usability and feasibility of the Healthcare version has not yet been assessed in clinical settings.

myfood24 is the first online 24 h dietary assessment tool that has been demonstrated to be suitable for use with different age groups in the UK [16]. Unlike traditional, paper-based dietary assessment methods, online tools are less costly and time consuming and have a lower burden on researchers. Some clinical populations, particularly those with conditions or diseases related to nutrition, may benefit from using this kind of technology to track their food intake. However, to date, very little research has looked at whether these new ways of assessing dietary intake can be successfully introduced into and used by clinical populations. A previous study by Gianfrancesco et al. [17] tested the usability of myfood24 in women with gestational diabetes and concluded that the tool could be used successfully as an online food record in clinical populations. However, response rates were low (121 of 199 (61%) completed a diary and 73 of 121 (60%) completed a survey), and the demographic characteristics of the participants in that study were very specific, reducing the representativeness of the sample; therefore, further testing of the usability of myfood24 in other clinical populations is needed. 

myfood24 Healthcare is available as an Android and iOS app for recording food intake and displays nutrient intake compared to standard United Kingdom (UK) dietary guidelines [18]. App access was provided without charge to HCPs and users during the study. HCPs did not receive incentives to recruit or assist participants. Future costs are expected to be covered by health provider networks and be free at the point of use for individual providers and patients. A demo of the myfood24 (online research version) can be viewed at www.myfood24.org (accessed on 5 April 2022). The Healthcare version of myfood24 used in this study differs from the original software in that patients can access the tool via a downloadable app on their smartphone or tablet. The software provides instant information on the nutritional content of foods and also provides analysis of nutrient intakes across any specified time period. For example, users can see daily or average nutrient intakes over a longer time, review the intake of different nutrients by meal events, and see the top foods that contributed to salt or fibre intake. To bring the food intake previously recorded on a particular day in line with nutritional recommendations, there is an optional feature to receive automated suggestions. These are generated using a linear programming ‘diet optimisation’ approach, and the app suggests foods to increase or add, or foods to reduce or remove from the diet. The automated suggestions aim to be as close to the user’s original diet as possible in order to minimise changes and therefore be acceptable. Diet optimization is a whole-diet approach that simultaneously combines data for different dimensions of diet improvement. A recent narrative review identified 67 published studies showing how mathematical diet optimization can help with understanding the relations between the different dimensions of diet [19]. However, none of those studies had used clinical populations to test the approach.

### 1.3. Study Objectives

We aimed to test the design of a randomised controlled trial (RCT) using myfood24 Healthcare in clinical populations in terms of recruitment procedures and intervention delivery, and to test the software usability and acceptability. More specifically, this study aimed to (i) quantify estimates for recruitment and compliance with the use of myfood24 Healthcare, (ii) examine the process and fidelity of intervention delivery for patients and HCPs, (iii) assess software acceptability and usability, (iv) quantify dietary intake in relation to targets, and (v) estimate cost-effectiveness. The wider goals were to gather user feedback in order to further develop myfood24 to better suit the users’ needs and to evaluate the feasibility of a larger study in which intervention effectiveness in relation to dietary and health outcomes could be assessed. Our hypothesis was that users would find the app acceptable and useful for tracking dietary data and providing HCP support to patients.

## 2. Materials and Methods

Two clinical settings were used to examine trial feasibility and the usability of myfood24 Healthcare in supporting patient care. The trial was registered in the ClinicalTrials.gov Protocol Registration and Results System, with reference ID: IRAS 266347. Patients were recruited either from the Gastroenterology Surgery Department at the Leeds Teaching Hospital NHS Trust or the Tier 3 Weight Management Services at York Teaching Hospital NHS Trust. Potential participants were identified from clinic lists and invited by their respective dietitian, nurse, or clinical team via e-mail or text message, and all participants gave informed consent to participate. The research team did not have access to the total number of participants approached since sensitive personal information had been retained by the clinical team who were responsible for inviting patients. Eligible patients included those aged 18 or over, receiving ongoing treatment in the department but not receiving palliative treatment for their condition, without any food allergies, eating disorders, or pre-existing conditions that would require a specific diet (except diabetes), able to read and understand English, and who have access to a smartphone or tablet computer with internet access.

Participants were randomised into 3 groups via minimisation and stratified into 2 groups each according to sex (male or female), age (years) (Weight management patients < 45 or ≥45; Gastrointestinal patients < 50 or ≥50), and body mass index (BMI) (kg/m^2^) (Weight management patients < 50 or ≥50; Gastrointestinal patients < 25 or ≥25). Group 1 received standard care, Group 2 was invited to use myfood24, and Group 3 was asked to use myfood24 with an automated feedback feature (details below). Groups 2 and 3 were asked to use myfood24 to measure their food and drink intake over at least 4 days during the study. The participants assigned to group 1 were offered the opportunity to use the myfood24 app at the end of the trial. The patients receiving standard care were invited to routine appointments with their HCP during the 2-month study but did not receive extra dietary guidance or support unless this was usual practice in the clinic.

HCPs added patient details onto the myfood24 Healthcare website to generate automated invitation e-mails; they were also able to review food and nutrient intakes and send notes to patients. Individualised nutrient intake goals (e.g., set a lower energy target) could be set by the HCP, or default settings with standard UK male and female intake recommendations could be used [18]. Participants that did not activate their accounts or begin to record their diets were contacted by their HCP or a researcher (DET) to check that the invitation e-mail had been received and that no difficulties had been encountered while accessing the app. During follow-up, some patients had phone or video consultations with their HCPs, providing an opportunity for HCPs to support patients in achieving their personalised dietary goals and improving the quality of their diet. 

Clinical notes were used to provide data on patients’ conditions or other clinical measures. Participants completed a feedback survey after 8 weeks, providing information on demographics, dietary requirements, previous experience of and attitudes towards the use of technology, and the usability of myfood24. Semi-structured interviews were conducted with HCPs to gain qualitative information of the usability of myfood24 in clinical populations. 

### 2.1. Automated Feedback

A linear programming ‘diet optimisation’ approach was used to analyse daily dietary intake and to compare this against standard or personalised nutrient targets for all nutrients. The app would then suggest dietary changes for participants. Those changes were presented as food items to add, increase, remove, or decrease in their diet. For instance, if a user logged an excessive intake of saturated fat and salt, an inadequate intake of fibre, and appropriate energy intake, the programme would comprehensively assess all foods consumed and tweak the diet to suggest the minimum number of changes needed to bring the diet in line with the targets. The suggested changes were designed to be based on the types of foods recorded in the diary, thus introducing minimal changes to the diet and making suggestions that would be more acceptable to the user. 

### 2.2. Patient Involvement

A patient advisory group (PAG) was established with patient representatives (PW, JG). Patients contributed to project design, reviewed participant-facing study documents, contributed to ethical submissions, and helped in software design. They had a complete overview of the app and tested it at two stages of its production, thus leading to improvements. The representatives were included in project meetings and assisted with the interpretation and presentation of the study results. All study participants were given the opportunity to receive information on the study results, and patient representatives assisted in preparing this feedback. 

### 2.3. Economic Analysis

In the absence of evidence of a clinical effect, a range of ‘what if’ assumptions were used with some data generated in this pilot study in order to create a number of economic scenarios based on the patients in the Tier 3 Weight Management Clinic. These scenarios showed what the potential economic impact could be if specific clinical effects were observed in a future trial of the tool. As well as data from the pilot, standard technology and deployment costs were used to generate the scenarios.

The Public Health England (PHE) weight management economic assessment tool (Available online: https://webarchive.nationalarchives.gov.uk/20170110165804/http://www.noo.org.uk/visualisation/economic_assessment_tool, accessed on 16 September 2021) was used to consider the economic impact of each scenario. The tool is designed to help decision makers assess the economic impact of existing or planned weight management interventions and to compare the costs of an intervention with potential cost savings. Based on changes in BMI, the tool compares the costs of the app with estimated health care cost savings, community-based social care cost savings, and the economic value of additional employment that would result from a reduction in disease incidence. It also calculates the cost per quality adjusted life year (QALY) gained from the use of the app. QALYs are recommended by the National Institute for Health and Care Excellence (NICE) as its preferred measure of health outcomes in technology appraisals. QALYs are a summary outcome measure used to quantify the effectiveness of an intervention, taking into account the quality and quantity of life gained. The cost in GBP per QALY is used to evaluate app utility.

### 2.4. Statistical Analysis

We anticipated that 300 patients would be eligible for the study during the recruitment period, with a 20% response rate, based on data collected from a small pilot. We therefore planned to recruit 60 participants from the Gastrointestinal Surgery Services in Leeds and 60 from the Tier 3 Weight Management Services, split into one site in Leeds and one in York. Staff at the Leeds Weight Management Service were redeployed in early 2021 due to COVID-19, and recruitment at this site could not proceed. 

Descriptive statistics were used to present details of rates of recruitment, attrition, compliance, and sample characteristics. All numerical data handling was performed using Stata 16.0 software. Interview data were analysed using a thematic analysis, which looked for potential themes within the data.

## 3. Results

### 3.1. Participant Characteristics and Feedback

In total, 48 participants consented and were recruited into the study: 21 patients from the Tier 3 Weight Management (T3WM) Services in York and 27 from the Gastroenterology Department in Leeds. A total of 16 were randomised to each of the three groups (usual care, myfood24 app, and app with automated suggestions). Survey questionnaires were filled in by 23 of 46 participants that completed the study (50%) (Figure 1). 

The majority of recruited patients were female (77%), and the average age was 51 (standard deviation (SD) 13) years. Weight management patients were younger on average (46 vs. 55 years) and had higher BMI (49.0 vs. 25.8 kg/m^2^) than gastroenterology patients (Table 1). Of the survey responses, 95% of participants were white, 68% were educated to at least A-level or equivalent, and 78% were non-vegetarian. 

Thirteen (59%) patients indicated that they have a medical condition that affects their diet, and 16 (70%) had received or were receiving ongoing dietary guidance from a healthcare professional. Six patients (23%) indicated they had not previously received dietary advice but would like to. Almost all (91%) had previously kept a record of their diet, and 9 (39%) had used an electronic device to record their diet in the past (Table 2). Participants were fairly familiar with using the internet, with 20 (91%) reporting usual daily use; the mean (SD) score for self-rated confidence in using technology was 8.2 (2.4) on a scale of 1–10, with 1 being not confident at all and 10 being extremely confident.

Compliance was reasonable among those assigned to test the app (n 32), with 25 (78%) using it at least once, and 16 (50%) recording intake over at least 4 days. Among the users, the mean (SD) days of use was 14.0 (17.5), and the median (interquartile range) was 6 (2.5 to 17) (range 1 to 58) during the 2-month intervention. HCPs at both sites elected not to modify the target energy and nutrient requirements for their patients, and these were therefore set to default male or female recommendations for adults (i.e., energy recommendations of 2500 kcal for men and 2000 kcal for women). The mean recorded daily energy intake was 1148 (SD 617) kcal (Table 3). The percent of total energy from total fat was notably lower, and the corresponding protein intake was higher in the weight management group as compared to the gastroenterology patients (29% vs. 37% for fat and 20% vs. 13% for protein). 

Of the 32 participants assigned to use the app, 16 (50%) had used the app and also completed the feedback survey. All but one participant reported using the app on their phone rather than a tablet computer, and the mean (SD) time taken to complete the food diary was 15 (10) min (Table 4). The participants were asked whether using myfood24 gave them confidence to stick to dietary advice from their HCP: four, four, and eight participants responded yes, no, and unsure, respectively. When asked if myfood24 could help to manage their conditions in the future, four, four, and eight participants responded yes, no, and unsure, respectively. When asked whether their conditions had improved as a result of using myfood24, three, ten, and three participants responded yes, no, and unsure, respectively. Even though the study follow-up was conducted only two months after, a third of the patients had already discussed diet as recorded in myfood24 with their HCP. Of those who responded, half said they would use the information to ask for dietary advice at their next clinic visit. 

The mean (SD) system usability score (SUS) was 59.2 (21.1) (median and interquartile range 57.5, 50.0–67.5) (Table 4). It was not possible to meaningfully quantify contextual factors associated with SUS due to small sample numbers. Questions on ease of use all scored above 2 (on a scale of 1–3), except for the ability to find the right food and add a recipe. Worthy of note is the fact that of those who responded, more than half wanted to use the feedback from myfood24 for a discussion with their HCP. 

Of the 16 participants assigned to group 3 with access to automated suggestions, seven either withdrew, never activated the app, or recorded their diet for less than 1 day. The remaining nine used the app for a mean of 18 (SD 22) days. Automated suggestions were activated on 88% of the days, a mean of 16 (SD 19) days of app use with suggestions. Example suggestions generated during the study can be viewed in Appendix A.

### 3.2. Health Care Professional Use and Feedback on myfood24

One HCP at each site reviewed online records for some of the patients and held video or phone consultations to discuss their diets. Nine consultations were held between the HCP and weight management patients, with the mean (SD) time taken to review the myfood24 records prior to the meeting being 11 min (SD 2.8), from a range of 5–15 min. The mean consultation length was 27 min (SD 7.8), from a range of 25–45 min. The average time taken to review records and hold consultations for five gastroenterology patients was 30 min. No safety concerns were noted during healthcare professional and patient consultations or at any other point in the study.

HCPs provided insight into their experiences with using the website and their views about the tool in general. There was agreement that the tool is useful for providing a rapid assessment of patients’ diets and is of potential value in supporting improved health outcomes. A couple of quotes from the HCPs: 

‘I have found myfood24 a really useful tool in supporting my Tier 3 weight management patients. The capability to review patients’ food diaries remotely ahead of a consultation helps to have a more tailored discussion about particular trends and identify any gaps in the diet. Patients have found using the app helps to keep them on track with their dietary goals’. 

‘For those using it and getting something out of it, it did help them to perhaps lose weight, or maintain weight, or even if their weight was not a particularly positive outcome from it, at least it was giving them more awareness and understanding of what they were doing…as a tool it is a beneficial thing for managing their condition’. 

The website helped to provide a quick summary of dietary data for use in patient consultations, particularly when patients had several days recorded. It is faster and easier than usual paper diaries. HCPs felt that the diets recorded were generally a good reflection of what was eaten. Regular app users said it increased their awareness and motivation to make healthier choices and stay on track. They felt there was a clear advantage to using the website for consultation and having all the data available for them to see. The website was easy to navigate and provided appropriate information, but some small changes in the layout were proposed by users to improve the app’s ease of use. HCPs agreed that further refinement of the app is necessary to improve usability and maintain the engagement of patients. For example, suggestions were made to improve the search function so users could more easily find the foods they consumed and to add a function for selecting items that are frequently consumed together. HCPs also agreed that while some patients had few issues while using the app and used it frequently, many patients could benefit from more training on how to effectively use the app and may benefit from additional ongoing support to maintain motivation and engagement. 

### 3.3. Process Evaluation

Recruitment was slower than anticipated, in part due to a change to remote recruitment because of COVID-19 restrictions. The lack of in-person contact at recruitment and not being able to demonstrate the features and benefits of the app in clinic meant that fewer patients than expected were interested in taking part in the study, which likely explains why compliance with using the app for a minimum of 4 days was moderate rather than good. Fewer patients than expected reported utilising the automated feedback suggestions feature in the app. Again, this likely resulted from ineffective communication with users about this feature, where to find it in the app, and the potential benefits that could be gained through having tailored dietary suggestions. 

The training delivered to HCPs was effective and appropriate, which meant that they could navigate the website and review patient records without difficulty. Instructions for patients were delivered by e-mail and in an electronically written user manual. It appears that the majority of patients did not read or refer to the user manual when they encountered difficulties with the app. 

### 3.4. Health Economic Estimates

The Public Health England weight management economic assessment tool was used to generate hypothetical scenarios to estimate the impact of reduced mean BMI due to technology on health and social care costs and patient outcomes; that is, if future research can show that the technology reduces mean BMI in the target population. There were no pilot data on BMI outcomes, but if the technology could be shown to contribute to an average reduction in BMI of 2 kg/m^2^ for a period of 1 year (from an average starting point of 44) in a weight loss population of 1000, the cumulative QALYs gained would be 96.8, with 0.2 premature deaths prevented over a 5-year period. Cumulative savings in health and social care costs during that period would be GBP 41,574. The cost–benefit ratio in the base case would show a positive return on investment after 3 or more years. Most of the benefit metrics demonstrate that more substantial impacts could be expected in the longer term (from 10 to 25 years) under this scenario, as might be expected from a primary prevention intervention. In this scenario, the cost per QALY gained is low for year one, which would indicate cost-effectiveness based on the NICE threshold of GBP 20,000 per QALY, with the intervention being cost-saving in subsequent years. These results are hypothetical, and it is important for future research to demonstrate that specified mean reductions in BMI across the patient population are attributable to the use of the technology. 

## 4. Discussion

This feasibility pilot RCT has demonstrated that the myfood24 Healthcare app can be used in clinical settings to support rapid dietary assessment and provide tailored support for patients from their HCPs. Furthermore, our findings demonstrate that the app is able to support patients in monitoring and modifying their own dietary intake through improved health literacy (i.e., knowledge about energy intake and intake of key macronutrients, salt, or fibre in relation to targets and the food sources of those nutrients). Important process evaluation outcomes have highlighted difficulties in recruitment and with fidelity of intervention delivery, providing essential insight for future applications of the tool in clinical settings. Patients and HCP feedback on the website and app usability have also informed future software development, which will ensure that users get the most benefit from the tool.

In 2020, 84% of UK adults on average owned a smartphone, spending 2 h and 34 min online on their smartphones every day [20]. The time it takes to complete a day’s intake using myfood24 (15 min in this study) would fit well into this general usage. The ease-of-use questions confirmed the general acceptability of the tool. However, the search and recipe creation functions were less well-regarded. 

This pilot trial was conducted during the strict lockdown situation in the UK resulting from COVID-19. This affected our ability to recruit patients; healthcare professionals were also not available to use myfood24 with their patients. We had planned to include a second Tier 3 Weight Management clinic in the trial, but the staff were all transferred to other duties. As part of the on boarding process, we wanted to demonstrate myfood24 to patients during a face-to-face clinic visit. These clinics were not in operation, so everything, including recruitment, was moved online. Feedback made it clear that although the app was easy to use, some patients would have benefitted from more support in using the system at the start. 

Compliance among app users (*n* = 32) was reasonable, with 25 (78%) using it at least once and 16 (50%) recording intake for four days or more. Among the users, the mean (SD) days used was 14.0 (17.5) and the median (interquartile range) was 6 (2.5–17.0) over 2 months. Previous studies have shown that recalling and recording food intake and receiving feedback are more likely to lead to better self-monitoring and changes in dietary intake [17,21]. This could ultimately have beneficial effects on clinical outcomes, as previously demonstrated for chronic renal disease and weight loss [22,23]. A relatively recent study using the weight loss app MyFitnessPal demonstrated that using the app alone is as good as other more intensive interventions [24]. We also found app use with MyMealMate (a forerunner of myfood24) to be effective for weight loss [23]. Patterns of self-monitoring are important, with a more frequent use of the app being a potential measure of success [25]. Adherence to self-monitoring can drop off over time; hence, reaching patients early in their treatment and maintenance phases via mobile prompts, virtual coaching, or other form of self-monitoring may help to retain interest and use [26].

In these two patient groups, the overall SUS was 59, lower than what we found in our test of the myfood24 online tool in women with gestational diabetes. Among these younger women, the SUS was rated good at 71 [17]. One explanation for this lower score was that the app version of myfood24 Healthcare was still under development at the start of the trial, and some participants may have experienced bugs which affected their use of the system. Identified bugs were fixed during the study. Two components of the SUS showed that myfood24 scored well: that participants thought the app was easy to use, and that people would be able to learn to use it quickly. 

Overall, the ease-of-use questions for the app scored better than average, with the feedback provided, time taken to use, and the ability to correct mistakes all scoring highly. However, participants could not always find the right food and drink items. The need for an effective search function has been highlighted for other diet apps under development [27]. myfood24 Healthcare app now has a database of over 89 thousand UK food items, including both branded and generic products, so an efficient search strategy is essential. myfood24 Healthcare does allow for searching by brand name or food category and also stores commonly consumed items to allow for quick selection by the user. Nevertheless, improvement is still possible. Aspects such as the search bar placement, hint text, and the way search results are displayed all contribute to how users engage with the search as well as the app as a whole.

Although the study was not powered to detect significant changes in clinical outcomes, of those patients who expressed their views, 3/16 patients (19%) felt that using myfood24 during the short study duration of 8 weeks had improved their symptoms/clinical condition. Apps have been shown to successfully support weight loss in other trials [3,23,25]. Mobile apps can facilitate weight loss through their ease of use and their ability to increase treatment adherence through self-monitoring. Weight loss and maintenance are achieved through high levels of engagement with a mobile app [25]. Long-term intermittent monitoring of diet using an app has been shown to be the most effective in delivering weight loss [28]. Participants who used the myfood24 Healthcare app tended to do so intermittently rather than every day; they were asked to use the app at least four times over two months. On the other hand, involuntary weight loss is frequent in advanced cancer patients, causing compromised anticancer treatment outcomes and function. The pathophysiology of cancer cachexia is characterized by negative protein and energy balance due to a variable degree of reduced food intake and deranged metabolism [29]. Protein intakes were low in the gastroenterology surgery patients, some of whom had been treated for cancer. If these patterns of intake were seen in a larger study, the impact of having better nutrition information for these patients could trigger a nutrition intervention and support improvement in longer-term outcomes. 

The low recorded energy intake in this sample reflects the weight loss diets undertaken by weight management patients, and that some gastroenterology surgery patients were receiving parenteral nutrition supplements or otherwise struggled to consume near-typical intakes owing to their medical and treatment history. Low recorded intake may also result from systematic underreporting. Although the total energy recorded was lower than the reference levels, this is not uncommon and is seen in other UK dietary surveys [30]. Additionally, the percentage of energy from total and saturated fat, protein, and carbohydrate are similar to that of the general population [31]. A large RCT of medical inpatients at nutritional risk showed that the use of individualised nutritional support to meet calorie and protein intakes improved important clinical outcomes, including survival, as compared with standard hospital food. This support included an HCP-led nutritional assessment and the introduction of individualised nutritional support in patients at risk [32]. That trial used regular daily food records assessed by a dietitian. The use of an app such as myfood24 would have simplified and sped up that process.

The use of online dietary assessment tools such as myfood24 in clinical populations is likely to be time-saving and cost-effective for health professionals in the long run as they offer a remote way of monitoring food intake and remove the need for in-person follow-ups. However, apps are currently under-utilised by healthcare professionals, including dietitians, in the nutrition care process [33]. This highlights a potential need for training and advocacy to enable a more effective engagement and the implementation of apps into practice in order to support patient education and behaviour change [33,34]. No other products similar to myfood24 have such an extensive, quality-checked product database, allowing patients greater accuracy to record actual intake. The app allows for instant nutritional analysis for any given time period, which can be viewed by patients and HCPs. The software also permits personalised nutritional goals to be set and tailored by the HCP according to patient needs. Although too few participants used the automated suggestions function in the app to be able to assess this aspect, it does provide further personalisation for the individual. 

The British Dietetic Association 6-step Model and Process for Nutrition and Dietetic Practice has Assessment as the first step. This requires the collection, analysis, and interpretation of relevant information to inform the dietetic intervention. Nutrition and food intake measurement is one of these important assessment processes [35]. Apps could provide support to dietitians to enable counselling and care and should not be considered a replacement for dietetic expertise [33]. Medical staff have long recognised the need for better nutrition training [36,37]. A new undergraduate curriculum in nutrition for medical doctors was launched in 2021 [38], which includes 13 core nutritional competencies divided into three categories: knowledge, assessment, and intervention. The myfood24 Healthcare app can provide Appendix A on patients’ food and nutrient intakes for HCPs; this would easily enable an assessment of a patient’s dietary intakes and provide suggestions for change through the optimisation function. 

Other apps that suggest dietary changes are available, such as the recently launched UK National Health Service Food Scanner app that suggests alternative healthier products [39]. However, this app only presents single food replacements, such as a snack that is lower in salt, and does not take the whole diet into account. By analysing the whole diet, myfood24 presents users with personally tailored suggestions, designed to address multiple nutrient targets at once. This approach ensures that suggested changes keep the user within all nutritional targets and may contribute to health literacy by demonstrating to users how to improve many aspects of the diet at once, with suggestions for foods to reduce/remove and increase/add into the diet. 

### 4.1. Strengths/Limitations

This is the first feasibility randomised controlled trial of the myfood24 Healthcare app used with patients and HCPs in a real-world clinical setting. Patients and HCPs provided feedback on the system with suggestions for improvements. myfood24 Healthcare is backed by validation studies, including comparison with nutrient biomarkers, and was developed with input from patients and HCPs. The underlying food composition database is extensive and quality-checked by nutritionists. Personalisation through the app is possible with the ability to amend nutrient targets, individualised feedback—viewable by patients and HCPs, and suggestions for improvement using diet optimisation. Two contrasting patient groups were included to provide a wider range of users than might be afforded by a group of individuals with a single health condition.

However, COVID-19 restrictions resulted in the termination of recruitment at one site, which meant in-person recruitment and demonstrations of the app were not possible, also contributing to lower recruitment than anticipated. COVID-19 restrictions also affected ongoing intervention delivery and follow-up assessments with HCPs. A further limitation is that the app was in the final stages of development when the study began, and some bug fixes were necessary in the early part of the trial. 

### 4.2. Next Steps

Given the need for personalised dietary support within many clinical settings [32] and the requirement for evidence-based tools [40], there is a strong case for examining clinical effectiveness in supporting longer-term changes, such as sustained weight loss and improved dietary habits, in a larger-scale trial. 

Future work beyond this feasibility pilot would benefit from a lengthier assessment of use in clinical settings to monitor and support dietary change and assess any consequent impact on clinical outcomes, including weight change or other markers of health status such as glycated haemoglobin, blood pressure, or blood cholesterol profile.

## 5. Conclusions

A multicentre randomised controlled trial in two distinct patient populations was undertaken to assess the feasibility of using the myfood24 Healthcare app as part of routine care in providing personalised dietary feedback and support. The study demonstrated that the app is acceptable for patients and HCPs, but more patient training and app updates are required. Our hypothesis was met in that patients and HCPs found the app software to be generally acceptable and beneficial for patients in tracking their dietary intake and assisting HCPs in supporting patients’ dietary changes.

Though this feasibility study was not powered to provide estimations of effect size, the outcome and process evaluation data support the feasibility of a larger clinical trial in which effectiveness could be quantified. Given the documented enthusiasm from HCPs and patients that used the system, there is a desire for technology such as this to support dietary assessment in clinical contexts.

## Figures and Tables

**Figure 1 nutrients-14-01768-f001:**
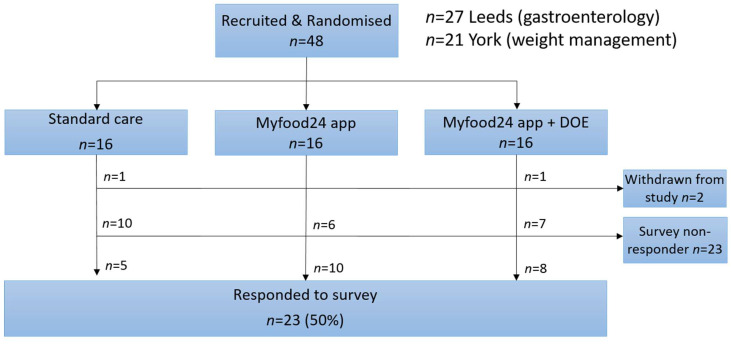
Flowchart of group allocation and survey response.

**Table 1 nutrients-14-01768-t001:** Participant characteristics at baseline.

	Tier 3 Weight Management Patients	Gastroenterology Surgery Patients	All
**Number recruited**	21	27	48
**Age, years**	46 (11)	55 (13)	51 (13)
**BMI, kg/m^2^**	49.0 (7.8)	25.8 (7.0)	35.9 (13.7)
**Sex**			
Male	3 (14)	8 (30)	11 (23)
Female	18 (86)	19 (70)	37 (77)
**Ethnicity ^1,2^**			
White	11	10	21
Black	0	0	0
Asian	0	1	1
**Highest educational achievement ^1^**			
Undergraduate, postgraduate, higher degree, or teaching qualification	3	8	11
AS or A-level/City and Guilds Technical or trade certificate	3	1	4
NVQ/GNVQ/CSE/O-level/GCSE	5	2	7
**Occupation ^1^**			
Managerial and Professional	3	4	7
Intermediate occupations	0	2	2
Semi-routine and Routine	3	0	3
Retired	2	5	7
Unemployed/non-paid work	3	0	3
**Usual eating pattern ^1^**			
Non-vegetarian	9	9	18
Vegetarian	2	1	3
Vegan	0	0	0
Other diet pattern	0	1	1

Values are mean (standard deviation) or *n* (%), unless otherwise stated. ^1^ Values are based on questionnaire responders only. ^2^ Some sub-categories were combined due to small numbers. Abbreviations: AS—Advanced Subsidiary level qualification, BMI—body mass index, CSE—Certificate of Secondary Education, GCSE—General Certificate of Secondary Education, GNVQ—General National Vocational Qualification, NVQ—National Vocational Qualification.

**Table 2 nutrients-14-01768-t002:** Prior dietary advice and technology readiness among survey respondents.

	Tier 3 Weight Management Patients	Gastroenterology Surgery Patients	All
**Questionnaire:** **Respondents/allocated, *n***	12/21	11/25	23/46
**Received dietary guidance or advice from HCP in the last 12 months**			
Yes, ongoing	9	3	12
Yes, once	2	2	4
No, but would like advice	1	5	6
No, did not want advice	0	1	1
**Dietary advice given by**			
Hospital doctor	2	1	3
Hospital nurse	2	0	2
GP	1	0	1
Practice nurse	0	0	0
Dietician	10	5	15
Private nutritionist	0	0	0
Other	1	1	2
Never received advice	0	4	4
**Conditions requiring dietary support**			
Obesity	12	0	12
Bowel cancer	0	1	1
Other cancer	0	0	0
Irritable bowel syndrome	1	1	2
Malnutrition	0	2	2
Other conditions	2	9	11
**Medical conditions or symptoms affecting diet**			
Yes	4	9	13
No	6	2	8
Not sure	1	0	1
**Have kept a paper or electronic food diary**	11	10	21
**Have kept a food record using computer, laptop, tablet, or smartphone app**	6	3	9
**Self-rated internet ability (score 1–5) ^1^**	Mean 2.4 (1.2)	Mean 1.8 (0.9)	2.2 (1.2)
**Internet use frequency**			
Daily	10	10	20
2–6 times a week	1	0	1
Less than once per week	0	1	1
Less than once per month	0	0	0
**Self-rated confidence in using technology in general (score 1–10) ^2^**	Mean 7.6 (2.5)	Mean 9.0 (1.3)	8.2 (2.4)

Note: Of the 48 participants recruited, 2 withdrew, and results therefore relate to the remaining 46. Values are mean (standard deviation) or *n* (%). ^1^ 1 = poor, 2 = fair, 3 = good, 4 = very good, 5 = excellent. ^2^ 1 = not confident at all and 10 = extremely confident.

**Table 3 nutrients-14-01768-t003:** Nutrient intake recorded using the app among app users.

		Tier 3 Weight Management Patients	Gastroenterology Surgery Patients	All
***N* allocated (groups 2 + 3)**		14	16	30
**Used app to record diet**		14	11	25
**Daily recorded nutrient intake**				
**Energy, kcal**		1060 (513)	1209 (675)	1148 (617)
Total fat	g/day	35.8 (22.9)	51.1 (33.3)	44.8 (30.4)
	% energy	29	37	34
Saturated fat	g/day	15.0 (10.8)	21.3 (21.3)	18.7 (18.0)
	% energy	13	15	14
Carbohydrate	g/day	133.1 (64.4)	141.9 (76.6)	138.3 (71.9)
	% energy	52	50	51
Total sugar	g/day	61.2 (40.4)	59.5 (37.2)	60.2 (38.5)
	% energy	25	22	23
Protein	g/day	50.3 (26.3)	42.5 (26.9)	45.7 (26.9)
	% energy	20	13	16
Fibre (AOAC) ^1^	g/day	15.4 (7.4)	9.7 (6.4)	12.0 (7.4)
Salt	g/day	3.9 (3.0)	4.0 (5.2)	4.0 (4.5)

Values are mean (SD) or % energy. ^1^ AOAC: Association of Analytical Chemists; their definition of fibre, including insoluble and soluble components.

**Table 4 nutrients-14-01768-t004:** myfood24 use and usability score in those using the app responding to the survey.

	Tier 3 Weight Management Patients	Gastroenterology Surgery Patients	All
**N allocated to app (groups 2 + 3)**	14	18	32
**Used app to record diet at least once**	14	11	25
**Number of food diaries submitted**			
**Mean (SD)**	10.3 (13.7)	18.9 (20.3)	14.0 (17.5)
**Median (IQR)**	5 (3–13)	8 (2–40)	6 (2.5–17)
**Range**	1–55	1–58	1–58
**Used app + completed survey ^1^**	9	7	16
**Device used to access app**			
Personal computer or Laptop	0	0	0
Smartphone	8	6	14
Tablet	0	1	1
**Time to complete 24-hour diary, minutes Mean (SD)**	16 (11)	13 (9)	15 (10)
**‘Did your s** **ymptoms/conditions improve as a result of using myfood24?’**			
Yes	2	1	3
No	5	5	10
Not sure	2	1	3
**‘Did you find all consumed food items when using myfood24?’**			
Yes	2	2	4
No	6	5	11
**‘Did you d** **iscuss diet, as recorded in myfood24, with HCP?’**			
Yes	6	1	7
Had a consultation but was not given dietary advice	0	0	0
No consultation since using myfood24	3	6	9
**‘** **If not, will you use myfood24 feedback to ask for dietary advice at next appointment?’**			
Yes	3	3	6
No	2	3	5
**‘Does myfood24 gives you confidence to stick to dietary advice from HCP?’**			
Yes	2	2	4
No	1	3	4
Not sure	6	2	8
**‘Could myfood24 help manage conditions/symptoms in the future?’**			
Yes	3	1	4
No	1	3	4
Not sure	5	3	8
**Likelihood of using myfood24 again (scale 1–10) ^2^**	5.6 (3.5)	4.6 (3.5)	5.1 (3.4)
**Ease of use (scale 1–3) ^3^**			
I thought the time taken to complete myfood24 was reasonable	2.4 (0.9)	2.3 (1.0)	2.4 (0.9)
I thought the instructions and wording used on myfood24 was clear and easy to understand	2.4 (0.9)	2.1 (0.9)	2.3 (0.9)
I liked the design and layout of myfood24	2.4 (0.9)	2.1 (0.9)	2.3 (0.9)
Finding the right food and drink items was simple and efficient	1.3 (0.7)	1.6 (0.8)	1.4 (0.7)
I found the selection of a portion size straightforward	2.3 (1.0)	2.3 (1.0)	2.3 (0.9)
I found the possibility to add a home-cooked recipe straightforward	1.8 (1.0)	1.6 (0.9)	1.7 (0.9)
If I made a mistake, I found it easy to correct	2.4 (0.9)	2.6 (0.8)	2.5 (0.8)
The feedback graphs were easy to understand	2.3 (1.0)	2.8 (0.4)	2.5 (0.8)
**System usability score ^4^ (Score 0–100)**	54 (17)	65 (26)	59 (21)
**Score components ^5^ (Score 0–4 with 2 being middle)**			
I think that I would like to use myfood24 frequently	2.0 (1.6)	2.3 (1.6)	2.1 (1.5)
I found myfood24 unnecessarily complex (it was complicated to use)	1.9 (1.4)	1.3 (1.4)	1.6 (1.4)
I thought myfood24 was easy to use	2.3 (1.2)	2.7 (1.3)	2.5 (1.2)
I think that I would need the support of a technical person to use myfood24	1.9 (1.1)	1.4 (1.4)	1.7 (1.2)
I found that the various functions in myfood24 were well-integrated (everything worked together smoothly)	2.0 (1.2)	2.1 (1.8)	2.1 (1.4)
**I thought there was too much inconsistency in myfood24**	1.9 (1.4)	1.9 (1.7)	1.9 (1.5)
I think most people would learn to use myfood24 very quickly	2.2 (1.1)	3.1 (0.7)	2.6 (1.0)
I found myfood24 very cumbersome to use	1.1 (0.9)	1.9 (1.9)	1.4 (1.4)
I felt very confident in using myfood24	1.9 (1.2)	3.1 (1.5)	2.4 (1.4)
I needed to learn a lot of things before I could get going with myfood24	1.9 (1.4)	0.9 (1.6	1.4 (1.5)

Values are mean (standard deviation) or *n* (%), unless otherwise stated. ^1^ All subsequent rows represent participants that used the app at least once. ^2^ Scale of 1–10, where 1 = not likely and 10 = extremely likely. ^3^ Disagree = 1, Neither agree nor disagree = 2, Agree = 3 and excluding ‘did not use feature’ responses. ^4^ Score out of 100 based on the 10-system usability score components. ^5^ Score of 0–4, where 0 = Strongly Disagree, 1 = Somewhat Disagree, 2 = Neither Agree nor Disagree, 3 = Somewhat Agree, 4 = Strongly agree. Abbreviations: %E—percent total energy, AOAC—Association of Official Analytical Chemists, HCP—health care professional, SUS—system usability score.

## Data Availability

Anonymised data presented in this study are available on request from the corresponding author.

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
