# Peer review of "Usability of myfood24 Healthcare and Mathematical Diet Optimisation in Clinical Populations: A Pilot Feasibility Randomised Controlled Trial"

_nutrients, 2022, doi:10.3390/nu14091768_

Round 1
Reviewer 1 Report
This randomized controlled study aimed to test ‘myfood24 Healthcare’, a dietary assessment app and healthcare professional website, in two clinical populations. The theme is interesting, but it is necessary for some reviews that I highlight below.
- Abstract: The aim of the study should be similar to the title “to test the usability of myfood24 Healthcare and mathematical diet optimization in clinical populations”
- Line 20: Insert the number of individuals recruited in each health service.
- Move the dot from before the references to after it in the entire manuscript.
- Lines 61-64: rewrite the sentence to be more clear. You are mixing gastroenterology and oncology patients.
- Line 64 – 65: describe the study better.
- Line 73: Where are the apps from?
- Line 87: Is this app free? Is HCP paid to guide participants through the app? Please include more information about the app.
- Line 93 and 106: Change myfood24 to Myfood24.
- Line 102: “However, response rates were low” – How many participants?
- Mention your study hypothesis in the introduction section and mention if your hypothesis was confirmed in the conclusion section.
- Line 142: how many patients were identified as a potential participants?
Line 149 – Did the participants sign the consent form? Mention the ethic committee approval.
- Line 228 – Did you recruit 48 patients or 48 patients accepted to participate?
- Line 231: Which questionnaires?
- Improve the title of Figure 1.
- Line 237: Change to 51±13 years.
- Table 1: All the acronyms must be mentioned after the table.
- Table 3: Fibre (AOAC)?
- Line 275 “15 (10) minutes”?
- The conclusions should summarize your main findings.
Thank you for the opportunity to review this manuscript!
Author Response
Please see attachment (includes responses to reviewer 1 and 2)

Reviewer 2 Report
This clinical trial assessed the feasibility of the myfood24 Healthcare app used with patients and HCPs. It is a novel and interesting study. However, it has a restricted sample size and a high loss rate during the trial (only 23 of 48 subjects completed the study). I suggest fixing all tables and results according to the final sample and modifying the title (to include "pilot study").
Minor comments:
Lines 151 - 152: According to the methods section, stratification of participants included BMI, confirm if weight management patients were stratified as stated in the manuscript: "... and body mass index (BMI) (Weight management patients <50 or ≥50 ...". If it is correct, please add units accordingly.
Table 2 shows n=46 subjects allocated to groups. However, in the manuscript, the total mentioned is n=48. Please correct.
Lines 449 - 450: "The low recorded energy intake in this sample reflects the weight loss diets undertaken". This assumption might be true, nevertheless, underreporting bias cannot be discarded, especially in obese subjects.
Author Response
Please see attachment - reviewer 2 comments are below those for reviewer 1.

Round 2
Reviewer 1 Report
The authors adequately revised the manuscript according to the requests. No further comments.
Author Response
Thank you
Reviewer 2 Report
The authors have adequately addressed most of my concerns in the revised version of the manuscript.
I have no further comments.
Author Response
Thank you